# Evaluation of Nano-Wall Material for Production of Novel Lyophilized-Probiotic Product

**DOI:** 10.3390/foods11193113

**Published:** 2022-10-06

**Authors:** Zin Myo Swe, Thapakorn Chumphon, Marutpong Panya, Kanjana Pangjit, Saran Promsai

**Affiliations:** 1Bioproduct Science Program, Department of Science, Faculty of Liberal Arts and Science, Kasetsart University, Kamphaeng Saen Campus, Nakhon Pathom 73140, Thailand; 2College of Medicine and Public Health, Ubon Ratchathani University, Ubon Ratchathani 34190, Thailand; 3Division of Microbiology, Department of Science, Faculty of Liberal Arts and Science, Kasetsart University, Kamphaeng Saen Campus, Nakhon Pathom 73140, Thailand

**Keywords:** gum arabic, cryoprotectant, freeze-drying, prebiotic, probiotic, synbiotic

## Abstract

Lyophilization is one of the most used methods for bacterial preservation. In this process, the cryoprotectant not only largely decreases cellular damage but also plays an important part in the conservation of viability during freeze-drying. This study investigated using cryoprotectant and a mixture of the cryoprotectant to maintain probiotic activity. Seven probiotic strains were considered: (*Limosilactobacillus reuteri* KUKPS6103; *Lacticaseibacillus rhamnosus* KUKPS6007; *Lacticaseibacillus paracasei* KUKPS6201; *Lactobacillus acidophilus* KUKPS6107; *Ligilactobacillus salivarius* KUKPS6202; *Bacillus coagulans* KPSTF02; *Saccharomyces cerevisiae* subsp. *boulardii* KUKPS6005) for the production of a multi-strain probiotic and the complex medium for the lyophilized synbiotic production. Cholesterol removal, antioxidant activity, biofilm formation and gamma aminobutyric acid (GABA) production of the probiotic strains were analyzed. The most biofilm formation occurred in *L. reuteri* KUKPS6103 and the least in *B. coagulans* KPSTF02. The multi-strain probiotic had the highest cholesterol removal. All the probiotic strains had GABA production that matched the standard of γ-aminobutyric acid. The lyophilized synbiotic product containing complex medium as a cryoprotectant and wall material retained a high viability of 7.53 × 10^8^ CFU/g (8.89 log CFU/g) after 8 weeks of storage. We found that the survival rate of the multi-strain probiotic after freeze-drying was 15.37% in the presence of complex medium that was used as high performing wall material. Our findings provided a new type of wall material that is safer and more effective and, can be extensively applied in relevant food applications.

## 1. Introduction

Gums are the natural dripples from the various parts of plants, especially the main trunk. When the stem is damaged (accidentally or purposely) by tapping, the injuries and beetle penetration result in the gum exuding from the bark. Gum arabic (GA) was noted “as a dried excretion got from the parts of the tree *Acacia senegal* (L. Willdenow) or related species of *Acacia*”, as defined by Joint FAO/WHO Expert Committee of Food Additives [1]. GA is non-sticky soluble fibers and is edible when sourced from, for example, dried gum, its exudates from the trunks and branches of *A. senegal* and *A. seyal* [2,3,4]. GA is widely used as it has excellent emulsifying properties, is non-toxic, and has low viscosity when compared to other polysaccharides. An emulsifier applied from GA is used in the pharmaceutical industry and in food plants. In addition, it is used as a stabilizer and suspending agent for insoluble drugs [5]. GA is the best emulsifier and stabilizer due to the binding of high water-solubility and low viscosity. *Acacia* gums are film-forming agents, stabilizers, thickeners, flocculants, emulsifiers and suspending agents. GA is an excellent thickener and suspender. These criteria underpin the standard and quality of GA utilized in the food, drug and cosmetic sectors [6]. Some scientists have studied the ability of GA as a nutrient for probiotic strains as it is fermentable fiber [7].

Probiotics are normal indigenous flora in the gastrointestinal tract, which can be prepared into various kinds of products. A probiotic is defined as “live microorganisms when it is administered in adequate amount gives a health benefit on the host” [8]. *Saccharomyces cerevisiae* and *Escherichia coli* microorganisms have also been used as probiotics, and *Lactobacillus* and *Bifidobacterium* species are commonly used as probiotics [9,10]. However, the strains used as probiotics are mainly in the group of lactic acid microorganisms. The benefits of a probiotic strain are explored in different kinds of foods. Before utilization, probiotic strains must survive and maintain their function during storage. Probiotic strains can survive storage a long time in a lyophilized product [11]. In the freeze-drying method, the solvent is freeze-dried and separated using sublimation. During the process, both the microbial cell wall and membrane can be injured by the transition of the membrane phase [11]. To avoid such issues, a protective wall agent is used to ensure the survivability of probiotic strains during the lyophilization process and storage [12].

There is pressure to increase the shelf life of drug products and foods. Thus, it is important to take care of their storage characteristics. The key element causing deterioration in the quality of merchandise is the water that is included in foods and pharmaceutical products. Thus, to eliminate the water from food and medicinal goods, a suitable drying technique should be used. Lyophilization, spray drying, and reduced-pressure drying are the most well-known drying techniques [13,14,15,16]. Lyophilization or freeze-drying has been investigated and used for over one thousand years to preserve biological cells [17]. Accordingly, the process of freeze-drying is frequently employed to produce dried viable probiotic products [18,19]. The aims of the present study were to assess the potential of probiotic strains and to produce lyophilized multi-strain probiotic product prior to investigation of the changes in the presence of protective wall materials during lyophilization and storage. GA, mung bean extract, stingless-bee honey and Riceberry extract were used as protective wall materials to assess their impact on the stability and viability of probiotic cells after freeze-drying and storage.

## 2. Materials and Methods

### 2.1. Materials

Gum arabic, mung bean, stingless-bee honey and a rice cultivar “Riceberry” were purchased from a local market in Thailand, which was certified according to Good Manufacturing Practice (GMP), Hazard Analysis and Critical Control Point (HACCP) and Good Agricultural Practice (GAP).

### 2.2. Growth of Probiotic Strains and Culture Conditions

Probiotic strains (*L. reuteri* KUKPS6103, *L. rhamnosus* KUKPS6007, *L. paracasei* KUKPS6201, *L acidophilus* KUKPS6107, *L. salivarius* KUKPS6202, *B. coagulans* KPSTF02, and *S. boulardii* KUKPS6005) were obtained from the culture collection of the Division of Microbiology, Faculty of Liberal Arts and Science, Kasetsart University, Kamphaeng Saen campus, Thailand. All the probiotic strains were grown on de Man Rogosa and Sharpe (MRS; Merck, Darmstadt, Germany) agar, nutrient agar (NA; Himedia, Mumbai, India) and yeast extract-malt extract agar (YMA; Himedia, Mumbai, India). All the probiotic strains were incubated at 37 °C for 24–48 h, stored in glycerol and kept for further investigation at −20 °C.

### 2.3. Determination of Cholesterol Removal Activity

The probiotic strains were investigated for their cholesterol removal activity. Thirty milligrams of cholesterol (polyoxyethanylcholesterol sebacate) (Sigma—Aldrich, Saint Louis, MO, USA) were dissolved in 10 mL of Milli-Q water prior to filter-sterilization using 0.45 µm filter (Millipore, Bedford, MA, USA) to obtain a stock solution of cholesterol. MRS, YM, nutrient broth (NB) and brain heart infusion (BHI; Himedia, Mumbai, India) broth containing 0.30% bile salt ox gall (Sigma—Aldrich, Saint Louis, MO, USA) were sterilized, and 1.5 µL/mL cholesterol stock solution was inoculated with 10^8^ colony forming units per milliliter (CFU/mL) of the activated probiotic strains (*L. reuteri* KUKPS6103, *L. rhamnosus* KUKPS6007, *L. paracasei* KUKPS6201, *L. acidophilus* KUKPS6107, *L. salivarius* KUKPS6202, *B. coagulans* KPSTF02, *S. boulardii* KUKPS6005 and mixed-culture of all probiotic strains) and incubated at 37 ℃ for 6, 12 and 24 h. Then, the solutions were centrifuged at 4000× *g* and 4 °C for 20 min. The cholesterol content in the supernatant was investigated using the method of Miremadi et al. [20]. A sample (1 mL) of the supernatant was mixed with 1 mL KOH (33%, *w*/*v*) (Univar, Ingleburn, NSW, Australia) and 2 mL 96% ethanol (Qrec, Auckland, New Zealand). The solution was vigorously shaken for 1 min and then incubated at 37 °C for 15 min. After that, the solution was cooled to 25 °C. Following cooling, 2 mL of Milli-Q water and 3 mL of hexane were added to the mixture, which was then vortexed for 1 min. The solution was left to settle until two distinct layers had formed. The upper hexane layer was collected and evaporated under nitrogen gas. Then, 2 mL O-phthalaldehyde reagent (Sigma–Aldrich, Saint Louis, MO, USA) were added and vigorously shaken for 1 min to dissolve the residues. After that, 0.5 mL of sulfuric acid (Qrec, Auckland, New Zealand) was added and vigorously shaken for 1 min and then rested at 25 °C for 10 min before the optical density was measured at 550 nm using a spectrophotometer (G10S UV-VIS; Thermo Scientific^®^, Carlsbad, CA, USA). The capacity of probiotic microorganisms to reduce cholesterol was expressed as the percentage of cholesterol removal at each incubation interval using following equation:Cholesterol removal percentage (%) = [1.5 − Residual cholesterol at each incubation interval/1.5] × 100 

### 2.4. DPPH Radical Scavenging Activity

The free radical scavenging activity of the probiotic supernatant was examined using the 1,1 diphenyl-2-picryl-hydrazyl (DPPH; Sigma–Aldrich, Saint Louis, MO, USA) method as described by Elansary et al. [21] with slight modifications. Briefly, an aliquot of 1.5 mL of stock solution of 0.1 mM DPPH in methanol was added to 0.5 mL of the sample solution (supernatant obtained from probiotic culture broth), with 1 mg/mL solution of gallic acid (Merck, Darmstadt, Germany) in methanol being used as the standard, followed by resting at room temperature in the dark for 30 min. Similarly, the same amount of methanol and DPPH was used as the control. The optical density at 517 nm was measured using a spectrophotometer (U-5100UV/VIS, Hitachi, Japan). An increase in DPPH radical scavenging activity is indicated by a decrease in the absorbance of DPPH solution. The experiment was carried out in triplicate. Free radical scavenging activity was calculated using the following formula:% DPPH radical-scavenging = [(*A*_control_ − *A*_sample_)]/*A*_control_ × 100
where, *A*_control_ = the absorbance of DPPH solution in the methanol
*A*_sample_ = the absorbance of DPPH solution with a sample or gallic acid solution.

The control contained 1.5 mL of DPPH solution and 0.5 mL of methanol.
Antioxidant activity = IC_50_ of gallic acid/IC_50_ of sample
where, IC_50_ is the inhibitory concentration at 50%.

### 2.5. Biofilm Formation of Probiotic Strain

Sterile test tubes were filled with 1 mL of MRS, NB or YM broth per tube. Twenty microliters of probiotic cells suspension were added to each tube and incubated at 37 °C for 3 days without agitation. Unbound probiotic cells were removed by washing the test tubes twice with sterile phosphate buffer saline solution (PBS; Himedia, Mumbai, India). Any probiotic cells that are surface-attached were stained with 0.1% (*w*/*v*) crystal violet in isopropanol-methanol-PBS (1:1:18 *v*/*v*) for 30 min, as described by Watnick and Kolter [22]. Excess dye was removed by washing the tubes three times with PBS. The surface-attached cells’ remaining dye was extracted using 1 mL of dimethyl sulphoxide. The extracted solution was measured using spectrophotometer (U-5100UV/VIS, Hitachi, Japan) at an optical density of 570 nm. The amount of crystal violet (in microgram) that adhered to the surface of the tube was calculated using the standard curve of crystal violet.

### 2.6. Screening of Gamma Aminobutyric Acid (GABA)-Producing Probiotic Strains Using Thin Layer Chromatography (TLC)

Probiotic microorganisms were inoculated in MRS, NB or YM broth containing 2% (*w*/*v*) monosodium glutamate (MSG) at 37 °C for 48 h. The cell-free supernatant was obtained by centrifuging at 9700× *g* and 4 °C for 10 min, which was screened for GABA using a thin layer chromatography (TLC) method. The GABA content in the cell-free supernatant was separated and identified using TLC on activated silica gel plates (Silica gel 60 F_254_, Merck, Darmstadt, Germany) according to the method [23] with a slight modification. The supernatant (1 µL) was spotted onto the TLC plates. Analytical standard γ-aminobutyric acid (Sigma–Aldrich, Saint Louis, MO, USA) was used as a positive control. The silica plates were placed in the mobile phase, including a solvent mixture of n-butanol: acetic acid: distilled water (5:3:2) and sprayed with 0.2% (*w*/*v*) of ninhydrin solution (Merck, Darmstadt, Germany). After that, the plates were heated at 60 °C for 30 min. Bands appeared on the TLC plates.

### 2.7. Production of Freeze-Dried Probiotic Product

#### 2.7.1. Preparation of Cell Pellets

Probiotic strains were inoculated in MRS, NB or YM broth for 24–48 h. All the probiotic strains were centrifuged at 3000× *g* at 4 °C for 10 min. The cell pellets were washed two times with 0.85% (*w*/*v*) sodium chloride solution (NaCl; Himedia, Mumbai, India) to remove the residue of the medium components.

#### 2.7.2. Preparation of Gum Arabic

30% (*w*/*v*) of GA was dissolved in distilled water and autoclaved at 121 °C for 15 min. The GA solution was kept cool at room temperature (25 °C).

#### 2.7.3. Preparation of Complex Medium

A sample of 10% (*w*/*v*) mung bean and 10% (*w*/*v*) Riceberry was heated using an autoclave (SS-245, Autoclave, Tomy, San Diego, CA, USA) at 100 °C for 20 min. After autoclaving, the extracted liquid medium was filtered using a 125 mm Whatman™ filter paper to clarify the medium. Then, 5% (*w*/*v*) stingless-bee honey was added to the medium and autoclaved at 121 °C for 15 min. Then, the medium was kept cool at room temperature (25 °C).

#### 2.7.4. Production of Lyophilized Product

The cell pellets of 7 probiotic strains were mixed together with the complex medium obtained from Section 2.7.3, which was also used as wall materials, at a ratio of 1:9 (cells to medium). After mixing the cell pellets and the wall materials, 2 mL of the mixture were put in tubes and frozen at −20 °C for 6 h. The mixtures were then freeze-dried using an Alpha 1-4 LSC basic freeze drier (Martin Christ, Osterode am Harz, Germany) at a condenser temperature of −55 °C and 0.128 mbar of chamber pressure. After 6 h, the lyophilized samples were kept within a sealed, airtight, plastic bag containing silica gel and stored at room temperature for 60 days. The lyophilized products were taken out of the packing at various storage periods to determine viability plate counts.

### 2.8. Probiotic Strains Adherence on Medium Using Scanning Electron Microscopy (SEM)

The lyophilized-samples were fixed to a sample slide and subjected to SEM. The medium adherence of probiotic strains was observed based on the method of Chumphon et al. [24]. SEM (MX-2000, MIRA3 scanning electron microscope, TESCAN^®^, Brno-Kohoutovice, Czech Republic) was used to determine the ability of the multi-strain probiotic to fasten to the medium (wall materials). Each sample was fixed with 2.5% glutaraldehyde and allowed to dry in a K850 Critical point drier (Emitech, Chelmsford, UK). Then, the sample was mounted on SEM micrograph stubs, coated with gold and observed using SEM. Scanning electron micrographs were observed at magnifications of 15,000×, 20,000×, 30,000× and 50,000×.

### 2.9. Determination of Cell Viability, Storage and Microbial Safety

All the lyophilized samples were stored at 25 °C and the viability of the probiotic strains was assessed at 7-day intervals over a period of 60 days. One gram of lyophilized samples was serially diluted with 0.85% (*w*/*v*) NaCl and poured on MRS, NA or YM agar. The agar plates were then incubated at 37 °C for 24–48 h and the number of cell colonies were counted. The probiotic cell counts were expressed as log number of CFU per gram of the lyophilized products. The viable cell counts were performed immediately after the process of lyophilization (day 0) and also at the end of the storage time (day 60). For microbial safety, the lyophilized samples were serially diluted from 10^−1^ to 10^−3^ with 0.85% (*w*/*v*) NaCl using the spread plate method. To check microbial contamination, eosin methylene agar was used for *E. coli*, BHI agar for total coliform count and YM agar for yeast and molds. The mean value of the viability of each of the probiotic strains was reported based on three plates. The survival rate of the probiotic strains in the treatment was expressed using the formula described by Savedboworn and Wanchaitanawong [25]:Survival rate (%) = *N*_1_/*N*_0_ × 100
where, *N*_1_ = the number of viable cells after the treatment (CFU/g) and *N*_0_ = the number of viable cells before the treatment (CFU/g).

### 2.10. Statistical Analysis

All experiments were recorded three times, with means and standard deviations calculated from these values. Data analysis was carried out using statistical analysis software (SSPS Inc., Chicago IL, USA, IBM^®^, Armonk, NY, USA). Analysis of variance was used to test significant differences between trials (*p* < 0.05).

## 3. Results and Discussion

### 3.1. Determination of Cholesterol Removal

A high blood cholesterol level is thought to be a threat for the cardiovascular disease [26]. Thus, decreasing the cholesterol level in serum is a main factor in preventing this disease. The cholesterol assimilation percentage was determined in vitro in the presence of 0.3% ox gall during anaerobic conditions at 37 °C. All seven probiotic strains and the multi-strain probiotic had different capacities to reduce cholesterol from the medium in the ranges 3.52–18.64% for 6 h, 17.53–38.58% for 12 h and 83.52–91.67% for 24 h (Figure 1). The strains *L. reuteri* KUKPS6103, *L. rhamnosus* KUKPS6007 and the multi-strain probiotic had significantly higher cholesterol removal activity than the other strains. Klaver et. al. [27] revealed that the removal of cholesterol by some lactobacillus strains was due to the disruption of cholesterol micelles that was caused by cholesterol deconjugation and precipitation with free bile salts as the pH media decreased due to acid production during the growth of lactobacilli. In addition, lactobacillus microorganisms are known for cholesterol assimilation, which is associated and incorporated in the cells during growth [28].

### 3.2. Antioxidant Activity: DPPH Radical Scavenging Assay

The antioxidant activities of the probiotic microorganisms are shown in Table 1. The DPPH assay is a free radical scavenging activity test that is stable at room temperature and is frequently employed to determine the antioxidant activities of hydrophilic molecules [29]. The strains *B. coagulans* KPSTF02, *L. rhamnosus* KUKPS6007 and *L. salivarius* KUKPS6202 had significantly higher antioxidant activity, compared with *L. reuteri* KUKPS6103 and *S. boulardii* KUKPS6005 based on the DPPH method. While *L. acidophilus* KUKPS6107 had the lowest antioxidant activity, *B. coagulans* KPSTF02 had the highest antioxidant activity of any of the other probiotic strains. During electron mitochondrial transport in aerobic respiration, the living microorganisms obtained an increasing level of oxygen radical by–products [30]. Numerous *Lactobacillus* microorganisms have hydrogen peroxide activity. The antioxidant enzymes of lactic acid bacteria, such as NADH oxidase, glutathione S-transferase, catalase, glutathione reductase and feruloyl esterase are neutralized by oxidative stress [31,32].

### 3.3. Biofilm Formation of Probiotic Strains

The biofilm formation capacity was strengthened in the probiotic strains, which were lyophilized with the medium (wall materials). The study measured the ability of biofilm formation based on the crystal violet fastened to biofilms produced by the probiotic strains after storage for 60 days (Table 1). The crystal violet binding was higher in *L. reuteri* KUKPS6103, *L. paracasei* KUKPS6201, *L. salivarius* KUKPS6202 and *L. acidophilus* KUKPS6107 4.61, 4.45, 3.82, and 3.74 µg, respectively. In contrast, the multi-strain probiotic, *L. rhamnosus* KUKPS6007, *S. boulardii* KUKPS6005 and *B. coagulans* KPSTF02 had lower crystal violet binding 3.36, 2.45, 1.25 and 0.98 µg, respectively, after storage for 60 days.

### 3.4. GABA-Producing Probiotic Strains Using TLC

Different probiotic strains were capable of producing GABA, showing the red spot that matched the GABA standard (Figure 2), after they had been screened using the TLC method. Seven probiotic strains and the multi-strain probiotic were screened for GABA production using 2% MSG. Other studies have revealed the highest GABA production after analysis of its culture supernatants using TLC [23,33,34]. Figure 2 represented the TLC profile of the cell supernatants obtained in the current experiment. Their refractive index (R_f_) values ranged from 0.60 to 0.66, which was similar to that of the GABA standard (R_f_ = 0.70). The prominent production of GABA was shown with L_3_ and L_4_ compared with the other sample strains after spot quantification. Thus, these probiotic strains were selected for future experiment.

### 3.5. Production of Lyophilized Probiotic Product

The present research showed that the viable cell count of probiotic strains was above 10^8^ CFU/g after a shelf life of 60 days. After storing for 30 days, the cell count of the samples steadily reduced a little. Similarly, the samples had lower cell numbers from 45 days until the end of the study (60 days). The survival rate of the multi-strain probiotic was 15.37% (7.53 ± 0.65 × 10^8^ CFU/g; Table 2) after the eighth week of storage. In addition, when compared with the standard lyophilized-products (control) that use skimmed milk as cryoprotective agent [35], the number of viable cells in control (standard lyophilized-products) was 6.53 ± 0.77 × 10^6^ CFU/g (0.19% survival rate). Thus, this reflects the fact that the potential of our wall material (complex medium) was high. In this research, it was found that the probiotic strains were suitable for the lyophilization product in combination with the complex medium (wall materials). All the probiotic strains survived well at room temperature until the end of storage. Therefore, the lyophilized samples combined with the complex medium are prone to being prominent nutraceutical products as a result of their good characteristics and the ability to maintain a constant viability.

### 3.6. Adhesion of Probiotic Cell as Determined Using SEM

SEM was applied to investigate the adhesion of the multi-strain probiotic to the wall materials. Strains survival was essential for encapsulation with the suitable medium (wall materials). The SEM observations of samples revealed the presence of probiotic strains with several morphologies that affected their ability to adhere to the medium (wall materials), as shown in Figure 3. The adhesion of probiotic strains was considered essential to achieve a beneficial probiotic cell effect in the gastrointestinal tract [36]. From the SEM studies, the evident adhesion showed that multi-strain probiotic was capable of adhering to the medium (wall materials). The probiotic adhesion was probably assisted by secreted lectin-like bacteriocins as a result of cell surface adhesion factors, such as lectin or the adhesion proteins of S-layers and lectin-like complexes on the probiotic cell walls [37]. The colonization of the pathogenic strains was restricted by the adhesion of the probiotic strains that were able to modulate the host immune system systematically [38,39]. Some mucous in the nanometer scale could be observed around the probiotic strains in the SEM micrographs (Figure 3f) due to the numerous wall materials. It was clear that the protectant wall materials were firmly adhered around the probiotic strains as a nano wall cryoprotectant based on their magnified images. From this experiment, the complex medium (wall materials) was essential for the survival and adhesion to the wall materials of the multi-strain probiotic.

### 3.7. Microbial Safety

The lyophilized samples were not contaminated by *E. coli*, yeast and molds after 60 days of storage, with the contamination being acceptable at lower than 1 × 10^1^–1 × 10^2^ CFU/g. Therefore, the lyophilized samples could be safely used for consumption in a functional product.

## 4. Conclusions

The multi-strain probiotic microorganisms showed high resistance to the lyophilization process in the presence of the cryoprotectants used. The best results were obtained when a complex medium (wall materials) was used. Synbiotic samples from lyophilized probiotic strains on the complex medium were formulated to achieve maximum probiotic viability. This study showed that the tested cryoprotectant materials influenced the viability of probiotic cells after the lyophilization processing. Furthermore, the multi-strain probiotic maintained a high degree of viability whilst in storage. The lyophilized multi-strain probiotics maintained their viability in the synbiotic product. Additionally, all probiotic strains were promising strains that produced many beneficial substances, and would be used as health supplements. The lyophilized synbiotic formulation provided a health benefit and had the additional effect of probiotic viability and stability. The mucilage around the probiotic strains was assumed as a potential nano-wall cryoprotectant based on their magnified SEM images. According to our key findings, this current study presented the distinguished development of lyophilized-synbiotic product in combination with multi-strain probiotic microorganisms and the new complex medium as nano-wall material.

## Figures and Tables

**Figure 1 foods-11-03113-f001:**
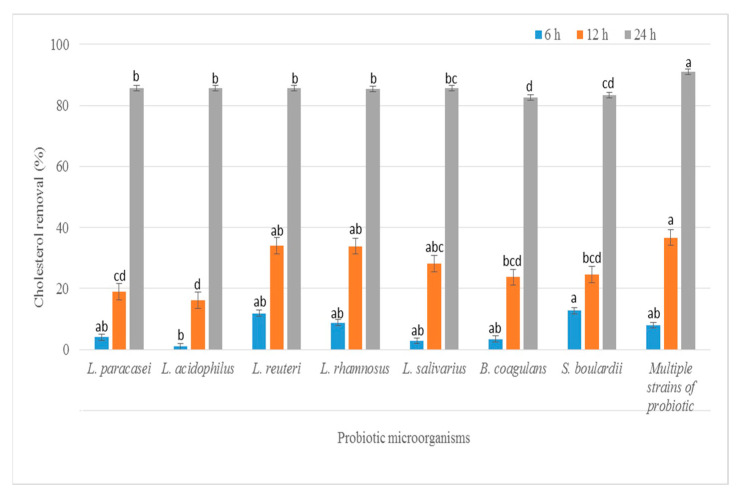
Cholesterol removal (%) of the seven probiotic strains and multi-strain probiotic incubated in media broth supplemented with 1.5 µg/mL water soluble cholesterol for 6, 12 and 24 h at 37 °C. Different lowercase between each time of incubation represents a significant difference using Duncan’s test with a confidence level of 95%.

**Figure 2 foods-11-03113-f002:**
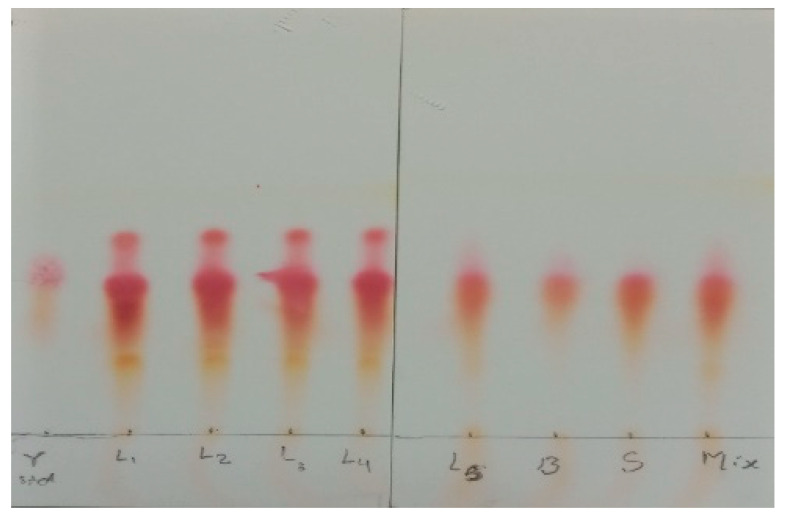
Chromatogram of screening GABA production by TLC method. The development solvent consists of n-butanol: acetic acid: water (5:3:2, *v*/*v*). The chromatogram was observed after spraying the plates with a 2% ninhydrin solution and developing at 105 °C for 5 min. (γ std: standard gamma aminobutyric acid, L_1_: *L. reuteri* KUKPS6103, L_2_: *L. rhamnosus* KUKPS6007, L_3_: *L. paracasei* KUKPS6201, L_4_: *L. acidophilus* KUKPS6107, L_5_: *L. salivarius* KUKPS6202, B: *B. coagulans* KPSTF02, S: *S. boulardii* KUKPS6005 and Mix: multi-strain).

**Figure 3 foods-11-03113-f003:**
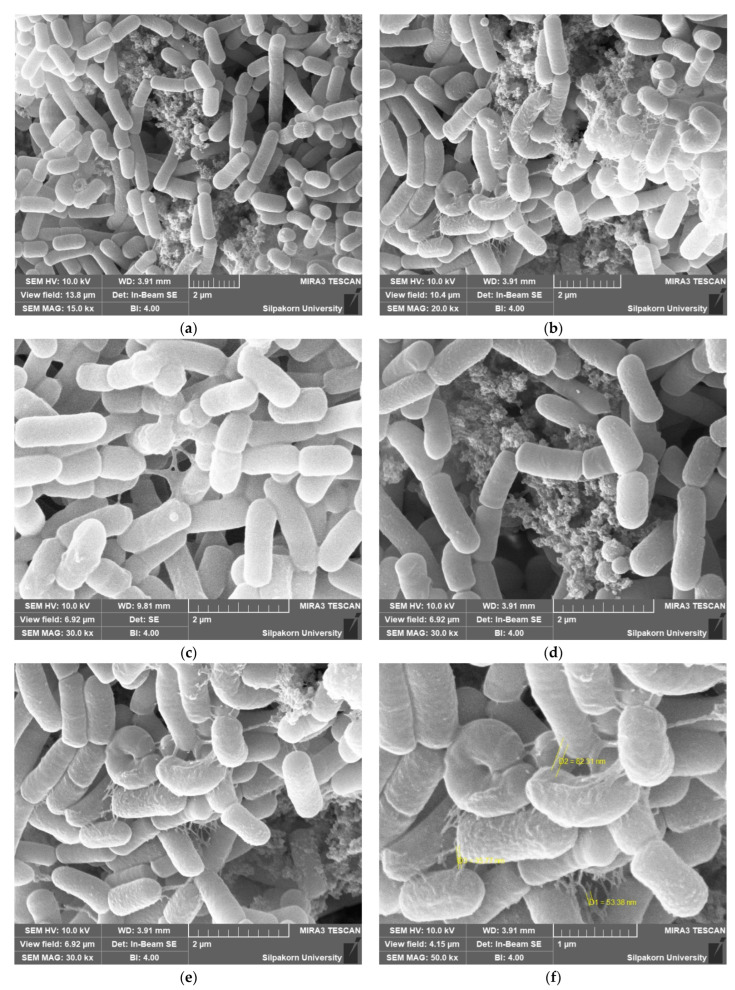
SEM of multi-strain probiotic product after freeze-drying in the presence of complex medium as protectant wall materials. (Magnification: (**a**) 15,000×, (**b**) 20,000×, (**c**–**e**) 30,000×, (**f**) 50,000×).

**Table 1 foods-11-03113-t001:** Biofilm formation and antioxidant activity of probiotic strains.

Strain	Amount of Crystal Violet (µg)	Antioxidant Activity (mg Gallic Acid/ mL Extract)
*L. reuteri* KUKPS6103	4.61 ± 1.17 ^c,^*	0.096 ± 0.041 ^a^
*L. rhamnosus* KUKPS6007 *L. paracasei* KUKPS6201 *L. acidophilus* KUKPS6107 *L. salivarius* KUKPS6202 *B. coagulans* KPSTF02 *S. boulardii* KUKPS6005 Multi-strain	2.45 ± 0.47 ^a,b,c^ 4.45 ± 0.02 ^c^ 3.74 ± 1.35 ^c^ 3.82 ± 1.48 ^c^ 0.98 ± 0.52 ^a^ 1.25 ± 0.20 ^a,b^ 3.36 ± 1.45 ^b,c^	2.246 ± 0.190 ^b^ 2.038 ± 0.219 ^b^ 1.893 ± 0.292 ^b^ 2.403 ± 0.298 ^b^ 0.220 ± 0.140 ^a^ 0.311 ± 0.090 ^a^ 2.310 ± 0.138 ^b^

* Values represent the mean ± standard deviation in three independent experiments, *n* = 3. A different lower case superscript letter in a column represents a significantly different value in statistics using Tukey’s test with a confidential level of 95%.

**Table 2 foods-11-03113-t002:** Survival rate of mixed-strains of probiotics in the lyophilized product after an eight-week period (at 25 °C).

Time (Week)	Viability of Cell Count (CFU/g)	Survival Rate (%)
0 1 2 3 4 5 6 7 8	4.90 ± 1.90 × 10^9^ * 8.26 ± 0.71 × 10^8^ 6.56 ± 0.78 × 10^9^ 1.75 ± 0.41 × 10^9^ 4.66 ± 0.86 × 10^9^ 3.76 ± 0.73 × 10^9^ 1.18 ± 0.08 × 10^9^ 9.56 ± 0.65 × 10^8^ 7.53 ± 0.65 × 10^8^	100.00 16.86 133.88 35.71 95.10 76.73 24.08 19.51 15.37

* Values show the mean ± SD of the three independent experiments.

## Data Availability

Data is contained within the article.

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
