# Peer review of "Evaluation of Nano-Wall Material for Production of Novel Lyophilized-Probiotic Product"

_foods, 2022, doi:10.3390/foods11193113_

Round 1
Reviewer 1 Report
Summary:
Line 27. CFU/g
Introduction:
Why is it only talking about gum arabic?
Methods:
Line 178. Value of room temperature
Why was honey not added at the same time as the other ingredients?
Line 189-190. Materials do not add up to 100%
Line 200. Does not mention the author
Results:
General: In this section it is not necessary to compare the results with references, for that there is the discussion section
Figure 1. In multiple probiotics, there is the letter a in the three bars, it seems that there were no differences between them, and so with the other strains. Then they must differentiate these results
What is expected by analyzing cholesterol removal at 6, 12 and 24 hours? Why was it not evaluated at 6-hour intervals?
Section 3.2. Table 1. Antioxidant activity should be expressed as IC50 and add its standard deviation.
Section 3.5. Table 2. Are there significant differences between the values of viability and survival rate?
Is there a way to determine adhesion quantitatively?
Section 3.7. The authors does not refer to any result (table or figure)
Supplementary materials needed
Reviewer 2 Report
Both the methodology and the results have been described inaccurately. Hence the work raises many doubts
The purpose of the work is not clear. If the effect of protective materials (actually one multi-component material) has been assessed, the results should be compared with the results for the material without protective substances. Neither the introduction nor the purpose of the study shows why the results from subsections 3.1 to 3.6 were presented. Do all results apply to lyophilized material? Were individual probiotics also freeze-dried?
No substantiation at work for the use of the term "nano"
7 probiotics and a multicomponent preparation were used for the research. Probiotics were incubated on 3 equal media: agar, nutrient agar and yeast extract-malt extract agar. Which of them were freeze-dried? Are they all? Have only microorganisms combined with the complex medium been lyophilized? If, as the title suggests, the goal was to evaluate a coated material, then experiments with the uncoated material should also be carried out.
On what basis was the composition of the complex medium developed?
What were the functions of the complex medium? The authors write in some places that it was wall material, in others they call it crioprotectant. If it was indeed a cryoprotectant (and thus protected against freezing, the material was subjected to vacuum drying, not freeze drying). Lyophilization and lyophilization are the same process !!
Line 105 The multi-strain probiotic composition must be specified
Line 127 What was sample solution?
2.7.4. The method of freezing the samples prior to lyophilization, which is essential for the survival of the microorganisms, should be described. There must be only frozen water in the lyophilized sample. Did the presence of the substance significantly lower the freezing temperature?
How practical are the results presented in Chapters 3.1 to 3.4?
Line 305. What future experiments were performed on the L3 and L4 strains?
3.5. Production of Lyophilized Product Development - title is unclear
Are the survival scores only for the mix? So why were single strain studies done?
Line 189-90. It is unclear whether the authors used one or two materials to protect the walls. The record The cell pellets were mixed with the medium (10% Riceberry, 10% mung bean, 5% honey and 30% GA), suggests there was one mixture.
It is not clear what Figure 3 Magnification shows: (3a) 15,000 ×, - one sample, (3b) 20,000 × - one sample; (3c-3d-3e-3f) 30,000 × - 4 samples, (3g-3h) 50,000 × 2 samples
Line 325-7. The authors write that: the lyophilized samples combined with the complex medium were the best nutraceutical products because of their good characteristics and the ability to maintain a constant viability. But compared to what were the best?
Line 355-7: Statement: From this experiment, the medium (wall materials) were the main factor in the survival and adhesion to the wall materials of the multi-strain probiotic. There is no justification, because studies with other factors have not been conducted
Line 370-2 In conclusion, the authors write: Furthermore, the wall materials in the freeze-drying medium protected against lethal destruction of probiotic strains much better than fructo-oligosaccharides. But they didn't research for the fructo ologosaccharides. Neither did they even report the results from the literature
Line: 375-7 Statement: Additionally, the synbiotics prepared using lyophilization could be formulated for use in probiotic food products to improve human wellness. The lyophilized synbiotic formulation provide a health benefit… it does not follow from the results presented
Line 378 - 380: The authors write: the mucilage around the probiotic strains was assumed as a potential nano-wall cryoprotectant based on their magnified SEM images. But what was the reason for such a statement
Round 2
Reviewer 2 Report
I accept it as it is
Author Response
Dear Reviewer,
Thank you very much for your approval and valuable comments.
Author